# Breeding Maize Maternal Haploid Inducers

**DOI:** 10.3390/plants9050614

**Published:** 2020-05-12

**Authors:** Henrique Uliana Trentin, Ursula K. Frei, Thomas Lübberstedt

**Affiliations:** Department of Agronomy, Iowa State University, Ames, IA 50011-1051, USA; ufrei@iastate.edu (U.K.F.); thomasl@iastate.edu (T.L.)

**Keywords:** maize maternal haploid inducers, doubled haploid technique, haploid seed production, haploid selection and verification, inducer cultivars, breeding approaches for inducer development

## Abstract

Maize doubled haploid (DH) lines are usually created in vivo, through crosses with maternal haploid inducers. These inducers have the inherent ability of generating seeds with haploid embryos when used to pollinate other genotypes. The resulting haploid plants are treated with a doubling agent and self-pollinated, producing completely homozygous seeds. This rapid method of inbred line production reduces the length of breeding cycles and, consequently, increases genetic gain. Such advantages explain the wide adoption of this technique by large, well-established maize breeding programs. However, a slower rate of adoption was observed in medium to small-scale breeding programs. The high price and/or lack of environmental adaptation of inducers available for licensing, or the poor performance of those free of cost, might explain why smaller operations did not take full advantage of this technique. The lack of adapted inducers is especially felt in tropical countries, where inducer breeding efforts are more recent. Therefore, defining optimal breeding approaches for inducer development could benefit many breeding programs which are in the process of adopting the DH technique. In this manuscript, we review traits important to maize maternal haploid inducers, explain their genetic basis, listing known genes and quantitative trait loci (QTL), and discuss different breeding approaches for inducer development. The performance of haploid inducers has an important impact on the cost of DH line production.

## 1. Introduction

The doubled haploid (DH) technology is the preferred method of inbred line development in many crop species [1]. Typically, maize (*Zea mays* L.) DH lines are created by crossing F_1_ or F_2_ plants with genotypes known as haploid inducers. Part of the seeds produced by these inducers have haploid embryos and are simply referred to as haploid seeds. When inducers are used as the pollen-source parent, they are called maternal haploid inducers, because only the genome of the maternal donor plant is transmitted to haploid embryos. When inducers are used as the seed-bearing parent, they are called paternal haploid inducers, because only the nuclear genome of the paternal donor plant is passed to haploid embryos. However, in contrast to maternal haploid inducers, paternal inducers also transmit their cytoplasmic genome to haploid seeds, which may or may not be desirable. Different genes control haploid induction in maternal and paternal inducers, and thus different inducers are employed depending on the method of haploid induction utilized [2,3,4,5].

The main application of paternal inducers is in the introduction of cytoplasmic male sterility (CMS) to inbred lines [6]. The first step in the conversion of inbred lines to a CMS background is to pollinate the paternal inducer with the inbred line to be converted. Haploid seeds are then selected, sown and the resulting plants pollinated by the same inbred line. This process restores the diploid state of the resulting embryos, while the male sterile cytoplasm is introgressed from the inducer. While paternal inducers with haploid induction rate (HIR) of up to 6% have been created [7], much more progress was achieved in maternal haploid inducers, where rates can exceed 15% [8]. HIR is calculated as the number of haploid seeds divided by the total number of seeds produced in a cross-pollination with a haploid inducer, and it is the common parameter to compare the haploid induction efficiency of different inducers. A frameshift mutation in a gene coding for a pollen-specific phospholipase, named *MATRILINEAL* (*MTL*)/*ZEA MAYS PHOSPHOLIPASE A1* (*ZmPLA1*)/*NOT LIKE DAD* (*NLD*), causes haploid induction in maternal inducers [3,4,5]. However, the HIR of maternal inducers is known to be under polygenic control [9,10,11,12,13]. In paternal inducers, the *indeterminate gametophyte* (*ig*) gene is responsible for haploid induction as a single gene trait [2].

## 2. Status Quo of Inducer Development

The first report of haploid plants in maize was concurrently made by L. F. Randolph and L. J. Stadler at the 1929 meeting of the American Association for the Advancement of Science [14]. In 1947, Chase [15] reported the natural occurrence of haploid plants in commercial inbred lines at frequencies lower than 0.1%. Researchers rapidly understood the potential that these haploid plants could have in maize breeding and started developing stocks with phenotypic markers that allowed the differentiation of haploid and diploid plants at different stages of plant development. They also observed that HIRs were affected by both the pollen-source and the seed-bearing parents and started breeding inducers for higher induction ability. In 1959, Coe reported a HIR of 3.2% in the self-pollinated progeny of inbred line Stock 6, which became the main germplasm source for the development of new inducers [16]. Haploid identification was greatly facilitated by the development of the Purple Embryo Marker (PEM) Stock [17]. This stock contains the *R1-nj* allele, which causes anthocyanin production in the aleurone layer and scutellum of seeds in which proper egg cell fertilization and zygote development occurred [17]. However, when improper fertilization of the egg cell or abnormal development of zygotic cells occur, the scutellum becomes unpigmented. This difference in scutellum pigmentation allows the differentiation of haploid and diploid seeds. Aleurone layer pigmentation allows differentiation of these two classes of seeds from outcrosses, which are unpigmented.

In 1969, Kermicle observed a 3% rate of androgenic (paternal) haploids in a strain of inbred line Wisconsin-23 (W23) [2]. Inducer breeding efforts continued mainly in Russia, where the Zarodishevy Marker Krasnodar (ZMK1) inducer-population (HIR 6–8%) was developed [18]. Direct selection within ZMK1 led to the development of ZMK1U, an inducer with HIR of 11–13% [19]. In France, the inducer line WS14 (HIR 3–5%) was developed from the cross between W23ig and Stock 6 [9]. Röber and colleagues [20] crossed WS14 with the Krasnodar Embryo Marker Synthetic (KEMS–HIR ~7%) to develop the German inducer RWS (HIR ~8%). In Moldova, Chalyk [21] developed the Moldovian Haploid Inducers (MHIs), which carried the *A1*, *B1*, *C1* and *R1-nj* alleles that jointly lead to anthocyanin production in different tissues (HIR 7–9%). In 2010, the Romanian company Procera reported the development of four inducers (PHIs 1–4), which contained all the anthocyanin marker alleles of MHI, *Pl1* and higher haploid induction rates (10–15%) [8]. *Pl1* leads to anthocyanin production in seedling roots and is a useful trait for the detection of false-positives after *R1-nj*-based selection.

CAUHOI, an inducer with high oil content (OC) and HIR ~3% was developed at China Agricultural University [22]. Xenia effects cause the OC of diploid seeds to be significantly higher than that of haploid seeds. This difference in OC can be detected by nuclear magnetic resonance (NMR), facilitating the automation of haploid selection. Additional breeding efforts in China lead to development of CAU5 and CAU079, inducers with both high OC and HIRs (HIR 6–8%) [23]. Efforts to automate haploid selection were also conducted in Germany, where high OC and HIR inducers were developed (UH600 and UH601-OC and HIR ~10%), and the efficacy of OC-based selection was evaluated [24,25]. The adoption of the DH-technology in tropical countries has not been as fast as in temperate countries, partly due to the lack of inducers adapted to tropical environments [26,27]. CIMMYT, in collaboration with the University of Hohenheim, developed two generations of tropically adapted inducers (TAILs & CIM2GTAILS-HIR 5–15%) [28,29]. In the United States of America, the Doubled Haploid Facility of Iowa State University (DHF-ISU) developed high HIR inducers carrying the *Pl1* and *Ga1* alleles (http://www.plantbreeding.iastate.edu/DHF/DHF.htm). *Ga1* confers unilateral cross-incompatibility, preventing fertilization by pollen grains that do not possess *Ga1*. This allele is typically employed in popcorn to prevent pollination by dent corn [30]. 

The majority of the abovementioned inducers are inbred lines, which were developed by the concomitant self-pollination of segregating families and evaluation of their haploid induction ability through crosses with different donors. Selection for qualitative traits, such as the anthocyanin marker genes, and for quantitative traits, such as HIR and tassel size, has traditionally been done phenotypically. The use of molecular markers for inducer development was only recently reported [31], despite of their low cost and usefulness in improving genetic gain [32,33,34,35]. However, phenotypic selection may still be the most cost-effective approach to select for traits that can be easily scored, such as purple embryo marker. Thus, clarification of the costs and benefits of different selection approaches for inducer development is still required and would be of great value for breeding programs adopting the DH technique or interested in improving the performance of the inducers used. The objectives of this paper are: (i) to compare the relative importance of different traits for maize maternal haploid inducers, (ii) to compare the advantages and disadvantages of different inducer variety types, and with this information, (iii) discuss the merit of different selection approaches for inducer improvement.

## 3. Genetic Basis and Putative Biological Mechanisms of Haploid Induction

It was recently discovered that a 4-bp insertion in the last exon of gene GRMZM2G471240, which encodes for a pollen-specific phospholipase, is required for the formation of seeds with haploid embryos in crosses with maternal inducers. The simultaneous discovery of this gene by different researchers rendered it different names: *MATRILINEAL* (*MTL*) [3], *ZEA MAYS PHOSPHOLIPASE A1* (*ZmPLA1*) [4] and *NOT LIKE DAD* (*NLD*) [5]. This gene is located within QTL *qhir1*, which explained up to 66% of the genetic variance observed in three populations derived from the cross of an inducer with distinct non-inducer lines [11]. Prigge et al. [11] observed strong segregation distortion against the *qhir1* locus and noted that while it affected the chance of a genotype having HIR > 0%, other QTL affected the mean HIR once HIR > 0%. Such observations indicate an epistatic control of HIR. A QTL located on chromosome 9, named *qhir8*, explained more than 20% of genetic variance in three different filial generations derived from the cross between inducers CAUHOI and UH400 [11]. A subsequent fine mapping study confirmed the positive influence of *qhir8* on induction rate when present along with *mtl*/*zmpla1*/*nld* [12]. It was recently reported that gene GRMZM2G465053 (named *ZmDMP*), which encodes for a DUF679 domain membrane protein, is the underlying causative allele of *qhir8* [13]. *ZmDMP* has the same subcellular localization and expression pattern as *MTL*/*ZmPLA1*/*NLD*, which is specifically expressed in membranes of sperm cells [5].

Single fertilization and the selective exclusion of inducer chromosomes from embryonic cells are two processes presumably involved with maize maternal haploid induction. In angiosperms, double fertilization occurs after precise and specific communication between pollen tube and synergid cell. Following their attachment and sperm cell activation, the egg and central cell fuse with the two sperm cells released by the pollen tube, resulting in a diploid embryo and triploid endosperm [36]. Single fertilization occurs when only the egg or the central cell are fertilized, creating kernels with defective endosperms or with haploid embryos, respectively [37]. Single fertilization events were observed in the Arabidopsis mutants *cdka-1*, *fbl17*, *msi1*, which occasionally produce pollen grains carrying a single sperm cell [38,39,40]. Arabidopsis and maize can compensate double fertilization failures by allowing the development of a second pollen tube, which is attracted by the second synergid cell [41,42,43,44,45,46]. This compensation mechanism is the origin of a process known as heterofertilization, which is established when the egg and central cell are fertilized by sperm cells of different pollen grains. The first report of heterofertilization in maize was made by Sprague in 1929, where it was reported to occur at an average frequency of less than 2.0% [41,42]. Substantially higher hetero-fertilization rates were observed when donors were first pollinated by an inducer and subsequently by a non-inducer, as opposed to initial pollination by noninducers followed by inducer pollination [23,47]. These observations suggest that single fertilization might be involved with haploid induction in maize.

More evidence exists for the process of selective exclusion of inducer chromosomes from maize embryonic cells [48,49,50,51]. Li et al. [48] observed that 43.2% of the haploids derived from crosses with inducer CAUHOI [22] carried a very small portion of the inducer’s genome (average 1.8%), and expressed inducer genes, such as *R1-nj* and genes affecting OC, during their development. Zhao et al. [49] detected that most inducer chromosomes were excluded from haploid embryonic cells within a week of pollination and from endosperm cells of defective kernels 15 days after pollination. Qiu et al. [50] reported that 7.4% of the haploid embryos derived from inducer HZI1 [52] carried segments of inducer chromosomes and noticed the presence of aneuploids, mixoploids, abnormal chromosomes and twin seedlings among its progeny. By introducing a CRISPR-Cas9 construct into a maize maternal haploid inducer and into an *Arabidopsis* transgenic line carrying a maize’s *CENTROMERIC HISTONE3* (*CENH3*) [53] transgene, which induces maternal and paternal haploids, Kelliher et al. [51] created genome edited haploid seeds in both species. Their findings provide evidence for the process of selective exclusion of inducer chromosomes from embryonic cells and demonstrate that CRISPR-Cas9 can be combined with different methods of haploid induction to effectively and rapidly edit monocotyledonous and dicotyledonous cultivars.

Environmental conditions at the moment of pollination, like temperature and moisture, were also reported to influence HIR [20,27,54,55]. Kebede et al. [27] reported higher HIR in winter than in summer in Mexico, while De La Fuente et al. [55] reported higher HIR in a warmer than in a cooler summer in Iowa. In barley, higher temperatures resulted in higher rates of chromosome elimination [56,57]. Silk age at the moment of pollination was reported to impact HIRs, with higher rates observed in older silks [58,59,60,61]. HIR is also influenced by the mode of haploid production, with higher rates being observed in hand-pollinations than in isolation fields [62]. Rotarenco and Mihailov [63] hypothesized that a higher heterofertilization frequency could be responsible for the lower HIRs observed in isolation fields. Since silks are exposed to pollen grains for a much longer period in isolation fields than in controlled-pollination nurseries, single fertilization events that might lead to haploid embryo formation could be compensated by double fertilization, thus decreasing HIRs.

The term inducibility was coined to describe the effect of donor germplasm on the relative frequency of haploid seeds [9,55,64,65,66]. Eder and Chalyk [65] observed HIRs between 2.7% to 8.0% when 20 different donors belonging to the flint, dent and flint x dent groups were pollinated by inducer MHI. HIRs ranging from 2.4% to 30.5% were observed in 30 hybrids created from a complete diallel of six inbred lines [55]. This complex nature of haploid induction, which is affected by environmental conditions and both inducer and donor genetic backgrounds, complicates inducer development by generating variability that is not solely attributed to inducer genetics. Despite of these challenges, HIRs were raised from 0.1% in the 1940s [15] to 14.5% in 2010 [8], and it is possible that the inducibility of the donor germplasm has been concomitantly improved as an indirect consequence of the utilization of the DH technique.

## 4. Breeding Goals in Developing Maize Maternal Haploid Inducers

The relative importance of different traits to inducers is context dependent, since there are multiple ways in which haploid seeds can be produced and selected. Whenever known, genes influencing these traits are mentioned. The best method to improve each trait depends on multiple factors, such as the availability of specialized equipment, human and financial resources, and to some extent, on the type of inducer cultivar used, namely inbreds, hybrids, or synthetics. The advantages and disadvantages of these three inducer types are compared and the merit of different approaches of selection for multiple traits are evaluated in the context of inducer development.

### 4.1. Marker Traits

#### 4.1.1. Color Traits

Most phenotypic markers used for the discrimination of haploid and diploid seeds are dominant anthocyanin markers, expressed in regular diploid or triploid tissues, but not in haploid embryos and tissues. Three genes (*C2* or *Whp1*, *A1* and *A2*) are required for anthocyanin production, and homozygous recessive genotypes for any of these genes do not synthesize anthocyanin. *Bz1* and *Bz2* are the subsequent genes in the anthocyanin production pathway, and the absence of any of them results in a reddish-brown pigmentation [67]. Additional genes control anthocyanin production in different plant tissues (Table 1). For a more complete review of the genes controlling anthocyanin production in different tissues, see [67]. Visual differentiation of haploid and diploid seeds today is primarily based on the purple embryo marker, which is encoded by the *R1-nj* transcription factor. The absence of the dominant allele interrupts the anthocyanin production pathway in the aleurone layer [68]. The time-consuming and laborious nature of visual haploid discrimination led scientists to investigate the possibility of automating this process. Boote et al. [69] achieved sorting accuracies above 80% by exploiting the *R1-nj* phenotype with fluorescence spectroscopy and imaging. Also exploring the pigmentation produced by *R1-nj*, De La Fuente et al. [70] were able to obtain accuracies greater than 50% in the majority of the genotypes evaluated with the multispectral Videometer3 system.

However, *R1-nj* expression is affected by multiple factors, such as environmental conditions and donor genetic background [20,27,28,71,72]. Moreover, seed morphology also affects the visibility of scutellum pigmentation. The scutellum of flat seeds, often found in the middle of ears, is much more visible than that of round seeds, often found at the top and bottom of ears. Moreover, mutations in genes that influence anthocyanin biosynthesis, such as *C1* and *C2*, negatively affect *R1-nj* expression [73,74,75]. The *Pl1* gene, which causes light-independent, anthocyanin production in seedling roots provides an additional mechanism to discriminate haploids and diploids that were misclassified based on the *R1-nj* marker. However, seedling roots of some genotypes can turn red when exposed to light, thus making classification based on the *Pl1* prone to errors. In conjunction with *Pl1*, the *B1* (or *R1-r* or *r1-r*) genes lead to anthocyanin production in seedling coleoptiles, leaf tips, margins and sheaths [67], aiding in the differentiation process. Plants homozygous dominant for *B1* and *Pl1* develop a dark purple pigmentation on husks and culm [67].

#### 4.1.2. Morphological Traits

Haploid seeds tend to have a lower mass than diploid seeds, partly due to their smaller oil mass and embryo size [76]. Lighter haploid seeds were observed in five out of six inbred lines tested by Smelser et al. [77], although the difference was statistically significant for only two of them. The lack of a substantial difference between the weight of haploid and diploid seeds impedes the exploration of this trait for seed discrimination. However, seed weight is a critical component for the accuracy of a fully automated nuclear magnetic resonance system proposed by Wang et al. [78]. Leaf morphology was also investigated as a method to differentiate haploid from diploid plants. Some genotypes, called disease lesion mimic mutants, develop symptoms that either mimic a disease infection or the resistance response associated with a disease infection. So far, over fifty lesion mimic loci, with dominant and recessive behavior, have been identified [79]. The symptoms displayed by each mutant are unique, and range from a few, small chlorotic lesions to lesions that cover the whole leaf [79]. The DHF-ISU developed inducers with a dominant lesion allele (*Les2*) [80]. Some of the issues encountered in this system were that the lesion phenotype was not always clear and its expression was light-dependent, as described by Hu et al. [81]. Plants fixed for the recessive *liguleless2* (*lg2*) allele are frequently used as donors in the development of inducer populations fixed for the domination version of this gene, since the presence/absence of ligule serves as an additional mechanism of haploid discrimination. However, this allele is rarely found in elite germplasm, and thus, it is not exploited by commercial breeding programs.

Adult haploid and diploid plants can be differentiated based on their vigor, leaf erectness and male fertility, characteristics that are part of the so called “gold standard” test [82]. Since the majority of inducers created to date are inbred lines, high heterosis is usually observed in the diploid progeny originated from the cross of an inbred inducer with a segregating population. The resulting diploid progeny tend to be more vigorous than the haploid progeny, with two-fold differences in plant and ear height commonly being observed. For example, Wu et al. [83] reported an average plant and ear height of 2.47 m and 1.44 m in diploid plants, and of 1.11 m and 0.46 m in haploid plants, respectively. Haploid plants have narrower, shorter and more erect leaves than diploid plants. Haploid plants also have significantly smaller stomata and guard cell length than crossed, diploid plants [84,85]. Haploidization is also associated with male sterility, and thus the presence or absence of extruded anthers is another indicator of a plant’s ploidy level. Although the differences between haploid and diploid plants become clearer at the adult stage, it is beneficial to discard diploid plants as soon as possible, in order to save time and resources during the genome doubling treatment and transplanting. Colchicine still is widely applied for genome doubling due to its ability of preventing microtubule formation during mitosis [86,87,88]. Despite of its carcinogenic properties, it is still one of the most effective molecules for doubling the genome of haploid plants [89].

#### 4.1.3. Oil Content

Xenia effects describe the chemical and morphological differences observed in seeds and fruits whose ovules were fertilized by genetically distinct pollen grains [90]. A classic example of a xenia effect is the multi-colored seed of the decorative Indian corn, which is generated by the independent assortment of multiple alleles affecting seed coloration. *R1-nj* and all of the aforementioned anthocyanin markers also display xenia effects. Embryo size is affected by the pollen genotype and is positively correlated with OC [91]. Approximately 85% of maize seed oil is found in the embryo [92,93], which accounts for about 12% of the total seed weight [94]. Multiple studies indicated that oil content is influenced by many genes with additive effect [95,96,97,98]. However, in crosses with high-oil haploid inducers, Melchinger et al. [24,25] reported that diploid embryos had OC levels below the average of the parents.

The first attempt to differentiate haploid and diploid seeds based on their OC differences was made at the Bavarian State Institute for Agronomy in 2002 [99]. The possibility of automating the selection process was soon envisioned, and inducers with high OC, such as CAUHOI, CAU1, CAU5, CAU79, UH600 and UH601, were developed [22,23,24,99]. As mentioned before, a fully automated screening system based on NMR was developed by Wang et al. [78]. OC is estimated by dividing the oil mass (OM) of each seed by its weight. Some factors like the OC of the donor parent and the introgression of inducer DNA segments into haploid embryos can affect the accuracy of seed discrimination based on OC [25,48]. Nevertheless, the possibility of automating haploid sorting through highly specialized equipment, promises to increase the efficiency and to reduce the cost of DH line production.

#### 4.1.4. Transgenic Approaches

Geiger et al. [100] investigated herbicide resistance as an alternative mechanism for haploid discrimination. These researchers incorporated a gene that confers resistance to phosphinotricin acetyl transferase into the German inducer RWS. Accurate differentiation of haploid and diploid seedlings was obtained by applying herbicide to the terminal portion of a leaf, where symptoms appear three to four days after spraying. While accurate, this approach cannot be easily adapted to a high-throughput platform because haploid plants die if thoroughly sprayed. Yu and Birchler [101] created the RWS-GFP inducer by introgressing a dominant green fluorescent protein (GFP) gene into RWS. They reported that high GFP expression in the endosperm of dry seeds hampered the inference of the presence of this protein on embryos. However, they noted that haploids could be easily discriminated by observing GFP expression in the radicles and coleoptiles of germinated seeds. With this approach, they were able to successfully discriminate haploids in five sweetcorn hybrids, germplasm in which *R1-nj* expression is commonly weakened by anthocyanin inhibitor genes. However, only thirteen among the eighteen putative haploids verified by chromosome counting were indeed haploids, indicating that this system needs further improvement. The necessity of specialized microscopes for fluorescence detection is another disadvantage of this system. Moreover, the possibility of introgression of transgenic DNA segments into DH lines [48,49,52,102] may discourage the utilization of transgenic inducers.

### 4.2. Agronomic Traits

#### 4.2.1. Plant Height and Lodging Tolerance

While there is no information in the literature on how different breeding companies produce haploid seeds, it seems reasonable to assume that well-established breeding programs, which produce a large number of DH lines every year, employ centralized isolation fields for haploid induction, whereas smaller maize breeding programs, which generate a limited number of DH lines annually, perform hand-pollinations in induction nurseries. Hybrid inducers may have a better performance than inbred inducers in isolation fields, due to their higher pollen production. On the other hand, shorter inducers with tassels at breast height may provide better ergonomics for induction nurseries.

Most of the haploid inducers available are inbred lines, which are shorter, less vigorous and more susceptible to diseases than hybrid plants. It is known that hybrid plants have larger tassels than inbred plants [103], and that tassel size is positively correlated with pollen production [104]. However, hybridization is associated with increased plant height, which in turn is negatively correlated to lodging tolerance [105]. Therefore, when employing hybridization as a method to increase pollen production, attention needs to be paid to lodging tolerance. Independent of the type of inducer utilized, shorter plants reduce the risk of lodging. Plant height is known to be a quantitative trait, and in maize, it is affected by genes with large effect displaying dominant and additive gene action [106,107,108,109,110,111]. Extensive genetic diversity exists for this trait, with plants as short as 0.7 m and as tall as 10.4 m being reported [112,113].

In a study designed to evaluate the percentage of genetically modified (GM) seeds harvested due to admixture with GM seeds at planting, Dietiker et al. [114] observed that the probability of fertilization of non-GM silks by pollen from GM tassels depends on the vertical distance between them. The shorter this distance is, the higher the chance of fertilization. This suggests that shorter inducers, which have tassels at the same height as the ears of donor plants, might be preferred for isolation fields. However, when observing pollen concentration profiles at different heights, Jarosz et al. [115] concluded that plant height only impacts pollen concentration at short distances. Maize pollen is one of the largest among anemophilous species (90–100 µm in diameter), usually settling nearby its source [116]. Thus, a possible disadvantage that taller inducers might have on pollinating silks of distant donor plants could be compensated by decreasing the ratio of donor to inducer plants.

Lodging tolerance is important for inducers, both in induction nurseries where hand-pollinations are performed, as well as in isolation fields. In inductions nurseries, inducers that lodge will force people to bend for collecting pollen, thus reducing ergonomics. In isolation fields, inducers prone to lodging are likely to generate a smaller seed set in donor plants, and consequently a smaller number of haploid seeds. Moreover, if hybrid inducers are employed on isolation fields, then hybrid vigor gained for plant height and tassel size might be accompanied by higher susceptibility to lodging. Since lodging tolerance is negatively correlated with plant height [105], breeding for both traits can be challenging. Nevertheless, QTL for maize stalk strength and plant height have been identified in different position of almost all maize chromosomes [110,117], and thus it should be possible to improve both traits simultaneously.

#### 4.2.2. Tassel Size

Whereas tassel size can be increased by heterosis through the process of hybridization, extensive genetic diversity exists for this trait, for which multiple genes have been identified [107,118,119,120,121,122,123,124,125,126]. Thus, genetic gain can be achieved even for inbred varieties. Fonseca [104] investigated how well different tassel morphologic traits correlate with pollen production in field conditions. He concluded that using a tassel area index, which integrated main stem diameter, main stem length and total branch length provided the best prediction of pollen production. However, measuring total branch length is laborious and time consuming.

Main stem length was the more robust and consistent morphological trait studied, explaining 67% of the variation in pollen production observed across years and genotypes. Since it also presents the advantage of being simple to measure, it can be used to estimate heritability and genetic gain for tassel size. Alternatively, utilizing an abstract score to evaluate tassel size may be the fastest way to improve this trait. The utility of high throughput phenotyping for tassel size estimation was already accessed [127]. Tassel size is negatively correlated with grain yield, and smaller tassels are observed in modern genotypes [128]. The simultaneous improvement of both traits is challenging since desirable alleles for each of them are not commonly found in high frequency at the same germplasm source.

#### 4.2.3. Pollen Production

The amount of pollen shed determines the efficiency in which haploid seeds are created both in isolation fields as well as in induction nurseries. Pollen production per plant, therefore, is an important trait for haploid inducers. There are multiple ways in which pollen production can be measured or estimated, and a trade-off exists between the accuracy of the method and its laboriousness. Vidal-Martinez [103] compared three methods to measure pollen production, namely, direct, cutting and bagging. He concluded that the last method, in which paper bags were placed over tassels daily to collect pollen, which was subsequently sieved and weighted, was the most effective. Fonseca et al. [129] was able to accurately assess pollen production by protecting tassels with clear bags (Pantek, Montesson, France) and quantifying the collected pollen through flow-cytometry. Although measuring pollen production provides a better estimate of inducers’ true pollen production capacity, it must be weighted by the additional efforts taken when compared to estimating pollen production based on tassel’s morphological characteristics.

#### 4.2.4. Length of Pollen Shed

The length of pollen shed likely impacts the efficiency in which haploid seeds are created. Both in isolation fields as well as in induction nurseries, multiple plantings of inducers are necessary to ensure pollen availability when donor plants start silking. If inducers with a longer period of pollen shed are employed, it may be possible to reduce one of the plantings of inducers, thus reducing costs and labor. The same benefits can be obtained if inducers with different maturities are available, but special attention should be taken to avoid long gaps between their pollen shed periods. Vidal-Martinez [103] observed that exotic germplasm produced almost twice as much pollen, over a period 50% as long, than adapted germplasm. These observations indicate that substantial genetic variability exists for this trait. Improving the tillering capacity of inducers is another way to extend their length of pollen shed, since tillers usually flower later than the main shoot. Interestingly, although selection for higher hybrid grain yield performed by commercial breeding programs lead to the reduction of plant height, tassel size and tillering capacity [128], inducer performance is increased by performing the opposite. Thus, employing older germplasm for inducer breeding may be advantageous.

#### 4.2.5. Seed Set and Tolerance to Ear Rots

A plant’s ability to set seed under self or open pollination determines the effort and cost necessary for inbred maintenance and hybrid seed production, respectively. Inducers often have low seed set, which can be partly attributed to the less vigorous germplasm used. Haploid induction ability is associated with embryo and endosperm abortion [23,50,130], thus reducing the amount of viable seeds after pollination. While the per se inbred performance has increased with breeding efforts, reduction in tassel size has also taken place, indicating a negative correlation between these two traits. Nevertheless, multiple genes affecting seed set have been identified [118,120,121,122,123,125,131,132,133,134,135,136], supporting the feasibility of improving both traits simultaneously.

Haploid inducers typically have small ears and are often prone to ear rots, what could be partly attributed to the old germplasm used in their breeding [137,138,139]. Moreover, the fact that most inducers available are inbred lines contributes to a generally reduced disease resistance. Many diseases, such as *Aspergillus flavus*, *Stenocarpella maydis* and *Fusarium graminearum*, can infect maize ears and kernels. Resistance to those diseases is controlled by multiple genes, and QTL associated with them already have been identified [140,141,142,143]. Therefore, different forms of marker-assisted selection (MAS), such as F_2_ enrichment and marker-assisted backcrossing (MABC), may be efficient ways to improve inducers by the incorporation of large effect QTL for disease resistance. Smaller effect QTL can be incorporated simultaneously or subsequently with phenotypic or genomic selection (GS).

## 5. Inducer Variety Types

The vast majority of available inducers are inbred lines. The only exceptions are the ZMK1 inducer-population [19], the German hybrid RWS/RWK-76 [54], and a tropically-adapted hybrid inducer developed by CYMMIT [155]. Here, we discuss the advantages and disadvantages of three kinds of inducer variety types (Table 2).

### 5.1. Inbred Inducers

Inbred inducers breed true and are uniform, facilitating maintenance and management. Their uniformity facilitates the identification and elimination of contaminations. Inbreds provide simplified logistics in comparison to hybrids by avoiding the concomitant maintenance of parental inbreds and hybrid seed production. If discrimination based on OC is desired, inbred inducers are probably the best type of variety to employ. Because OC-based discrimination requires the determination of thresholds, differences in OC levels of inbreds used to form a synthetic population or a hybrid could lead to classification errors. Most disadvantages of inbred inducers are associated with inbreeding depression, such as reduced vigor, smaller morphological characteristics and higher susceptibility to diseases. A drastic reduction in seed set might also be observed as a consequence of high homozygosity. Inbred inducers produce less pollen than hybrids inducers, and thus, are likely to have a weaker performance in isolation fields. However, they may provide better ergonomics for induction nurseries, where hand-pollinations are performed.

### 5.2. Hybrid Inducers

The process of hybridization can easily and simultaneously improve multiple traits important to inducers, such as tassel size, pollen production and disease resistance. No hybrid vigor has been observed for HIR [155], consistent with the fact that this is a gametophytic trait. A challenge in employing hybrid inducers is that both parents must be fixed for the same marker traits, otherwise differentiation of haploid and diploid plants based on those characteristics becomes unreliable. Another challenge commonly associated with hybrid seed production is the creation and maintenance of separate genetic pools. Moreover, employing hybrid inducers requires the continuous production of both hybrid and parental seed. Since hybrids are taller than inbreds and synthetics, more attention has to be paid for lodging tolerance [105]. If discrimination of haploid and diploid seeds is based on OC, two genetically distinct parents with similarly high OC levels need to be developed to ensure accurate seed discrimination when hybrid inducers are be employed. Developing such parents would be both challenging and time-consuming. However, if discrimination is based on *R1-nj*, all inducer types are equally suitable.

### 5.3. Synthetic Inducers

Synthetic inducers can combine the advantages of inbred and hybrid inducers. Hybrid vigor is recovered to some extent in synthetics, depending on the genetic dissimilarity between the founder genotypes used to establish them. Synthetic populations are easier to maintain when compared to hybrid inducers, since they need to be reestablished less frequently. Synthetic inducers might shed pollen for longer periods of time when compared to both inbred and hybrid inducers, due to the genetic variability found in them. However, synthetic populations do not exploit heterosis as well as hybrids do, and thus are more susceptible to diseases and produce less pollen. Synthetic populations need to be renewed periodically by recombining the original parents, due to loss of vigor and desired characteristics generated by drift. This implicates that parental seed needs to be saved and increased regularly, which is a disadvantage when compared to inbred lines. Moreover, the fixation of marker traits is more complicated in a synthetic population with multiple parents than in a single inbred line.

## 6. Breeding Procedures and Strategies for Inducer Development

Independent of the breeding strategy, a certain sequence of selection steps is recommended in haploid inducer development. To describe a typical breeding approach for inducer development, we propose the scenario of a tropical breeding program interested in developing its own, adapted inducer from a licensed, exotic inducer. The exotic inducer is fixed for *R1-nj*, *Pl1*, *mtl* and *zmdmp*, but has poor agronomic performance and environmental adaptation. This inducer will be crossed to a set of elite inbred lines which possess good agronomic performance and environmental adaptation, but lack the desirable alleles at these four genes (Table 3). Since genes that hamper *R1-nj* expression occur at a higher frequency in tropical germplasm [72], employing a set of adapted inbreds, rather than only one, reduces the chance of developing inducers with poor *R1-nj* pigmentation. Segregation for *R1-nj* will be observed in F_2_ seeds, where seeds displaying anthocyanin pigmentation in the scutellum and aleurone layer have at least one copy of this allele and can be discriminated from homozygous recessive seeds, which are uncolored. From our experience, it takes approximately 15 minutes for a person to discriminate an ear with 400 seeds using the *R1-nj* phenotype. Assuming a wage of $12.00 per hour, the cost per seed selected would be $0.0075. If available, a color sorter machine color can be used to rapidly separate colored from uncolored seed at a very low cost. If early fixation of *R1-nj* is a goal, then MAS can be applied on pre-selected F_2_ seeds. The price to outsource MAS with a single simple sequence repeat (SSR) marker is roughly U$1 per plant [156,157]. DNA extraction accounts for approximately 16% of this cost [157], which reduces part of the cost per additional marker employed. In well-stablished breeding companies, in house genotyping costs are usually around $0.10 per data point per genotype. If seed chipping and high-density genotyping are available at a low cost, then MAS or GS can be applied to reduce the number of F_2_ individuals to be tested in the field.

F_2_ plants that were self-pollinated and are homozygous for *R1-nj* can be easily identified and selected at the moment of harvest, since all of their seeds should show the typical anthocyanin pigmentation produced by this allele. The red root marker, conditioned by the dominant *Pl1* allele, can also be visually selected. However, its selection is more laborious than the purple embryo marker, since seeds need to be first germinated in small trays and, after selection, transplanted to either bigger containers or the field. Employing markers to select for this allele allows direct sowing in the field, simplifying operations and reducing costs. The presence of the *mtl* and *zmdmp* alleles, which disrupts maternal haploid induction [3] and increase the HIR of inducers fixed for *mtl* [13], respectively, can be verified by progeny testing. However, MAS is likely the easiest way to select for the presence of these alleles. Because strong gametic and zygotic selection against *mtl* occur [11,23,158], self-pollinating genotypes heterozygous at this locus will not produce the expected frequency of homozygous recessive individuals. From our experience, only 12–15% of F_2_ plants are fixed for the *mtl* allele [80]. Therefore, the rapid fixation of the *mtl* allele is desirable to simplify the selection process.

Just a small fraction of the F_2_ plants (~0.4%) will be fixed for *R1-nj*, *Pl1*, *mtl* and *zmdmp*. An extremely large population of F_2_ plants would have to be sown if a reasonable number genotypes fixed at these loci were to be selected. Since *mtl* is required for haploid induction and *R1-nj* can be inexpensively selected, MAS in the F_2_ generation can be used to ensure the fixation of *mtl* and elimination of genotypes homozygous for the *pl1* or *ZmDMP* alleles. This will increase the chance of obtaining F_3_ genotypes fixed for *R1-nj*, *Pl1* and *zmdmp*. In this generation, phenotypic selection can be used to fix *R1-nj*, while MAS can used to fix *Pl1* and *zmdmp*. Table 4 lays out a breeding scheme for inducer development.

Plant height, lodging tolerance, tassel size, pollen production, length of pollen shed, HIR, seed set and tolerance to ear rots are traits that affect inducer performance and are under polygenic control. Different breeding methods, such as phenotypic selection, MAS, and more recently, genomic selection, have been used to improve quantitatively inherited traits. Genomic selection requires genotyping and phenotyping a panel of individuals to establish a prediction model with which to estimate the performance of individuals that were only genotyped [159]. Despite requiring an additional initial effort for model construction, its value for the improvement of quantitively inherited characteristics has already been demonstrated [35,160,161,162], and is now widely adopted in commercial breeding programs. Employing genomic selection for the simultaneous improvement of all polygenic traits important to inducers might be economically interesting, since the cost associated with phenotyping a large sample of individuals is presumably high. In addition, genotyping costs continue to decrease [163,164].

The value of genomic prediction for the improvement of traits important to haploid inducers was already assessed, and moderate to high prediction accuracies were observed in most of them [35]. For polygenic traits affected by large effect QTL, such as plant height, flowering time and HIR, a combination of *de novo* GWAS and GS might result in higher prediction accuracies than those observed in GS [165]. In the *de novo* GWAS + GS approach, markers with lowest p-values are treated as fixed effects in the prediction model, whereas in GS, all markers are treated as random effects [165].

Evaluating HIR is time-consuming and labor-intensive, since usually many cross-pollinations are performed to produce enough seeds with which to obtain a reliable estimate of a genotype’s induction ability. Visual discrimination of haploid and diploid seeds is also very time-consuming, besides being quite susceptible to human errors, which affect the quality of the data collected. For instance, multiple factors can affect *R1-nj* expression and its visibility in the scutellum, such as environmental conditions [20,27,28,71], seed morphology and the presence of inhibitors genes in the donor germplasm [73,74,75]. These factors complicate haploid discrimination, which when coupled with a natural variation in selection stringency among evaluators, might affect the reliability of the estimates of HIR. Therefore, when evaluating the HIR of different inducers for the purpose of selection, a good balance needs to be found between sample size and the number of people involved in this task. Because of the high cost, labor-intensive and complicated task of obtaining reliable estimates of HIR, genomic selection might be helpful to improve this trait.

Tandem selection, independent culling, and index selection are common approaches used for the selection of multiple traits. Tandem selection proposes improving one trait at a time until the desired level for a given trait is achieved. This strategy is useful when selection for different traits occurs during different stages of a breeding cycle. For instance, in the first generations of an inducer breeding cycle (e.g., F_2_–F_3_), selection for qualitative traits, such as *R1-nj*, *Pl1* and *mtl*, or traits with moderate to high heritability, such as flowering time and plant height, can be effectively performed. At later stages of a breeding cycle (e.g., after F_4_) selection for low heritable traits, such as yield and resistance to some diseases, is more effective.

The strategy of independent culling levels proposes establishment of thresholds for each trait, and only genotypes that reach those thresholds are selected. For instance, if a breeder decides that an inducer must produce at least of 200 seeds in cross-pollinations and that it must have an HIR above 5.0% to ensure the economic viability of the DH technique, then all inducers that do not meet both criteria are discarded. The disadvantage of this selection strategy is that genotypes with very good performance for one trait, but with a below-threshold performance for another, are discarded, even though they might still have value as a source of favorable alleles for a trait.

A way to avoid the strictness caused by thresholds, and to allow compensation among traits, is to employ index selection. In index selection, the phenotypic value that each genotype has for a trait is multiplied by the weight given for this trait. All weighted phenotypic values are accumulated in the index value for that genotype. The weight might be the economic value for that trait, or a value empirically determined by the breeder. Different kinds of selection indices exist, and their usefulness is context dependent. A challenge associated with selection indices is to determine appropriate weights for each trait, since they depend on the unit of measure of each trait and because they might not be stable (e.g., prices might change from season to season).

For inducer development, all methods are used and embedded in a flexible tandem selection scheme (Table 4). In the F_2_ and F_3_ generations, phenotypic or MAS are used to fix *R-nj*, *Pl1*, *mtl* and *zmdmp*. Then, culling is used to discard F_3_ families with very low HIRs. Finally, index selection is used to identify F_4_ or later generation families having the best combination of HIR and various agronomic traits. In particular for this last stage of selection, GS is a promising alternative [35], since prediction accuracies for HIR and the other traits important to inducers were found to be high, and their phenotypic evaluation is laborious and thus costly.

## 7. Conclusions

In vivo haploid induction through crosses with maize maternal haploid inducers became the main method of DH line production in well-established breeding programs. However, smaller breeding operations still were not able to take full advantage of this rapid method of inbred line production. One of the main reasons for this lag is the lack of adapted or efficient haploid inducers, which is frequently observed in tropical countries, where inducers breeding efforts are more recent. Clearly defining traits of importance, breeding approaches and the most appropriate type of inducer cultivar for each breeding program is a fundamental step in a maize breeding program interested in implementing the in vivo DH technique. In this review manuscript, we explain each of these topics in detail. In addition, we synthesize the history of haploid inducer development, mentioning the most important cultivars created. This might be useful for the determination of which sources can be used for the development of inducers adapted to the environmental conditions of each breeding program. We hope that this review manuscript will help in the development of more efficient inducers, in the expansion of the maize in vivo DH technique and in the consequent reduction of cost of hybrid seed production.

## Figures and Tables

**Table 1 plants-09-00614-t001:** List of different traits important to haploid inducers, along with their genetic control, known genes or QTL, mode of gene action, observed trait range, breeding goal and brief explanation of their importance.

Trait	Genetic Control	Known Genes or QTLs	Gene Action	Trait Range	Breeding Goal	Why It Is Desirable	Reference
Purple embryo marker	Monogenic	*R1-nj*	Dominant		Fixation	Haploid selection at the seed stage	[17]
Red root	Monogenic	*Pl1*	Dominant		Fixation	Haploid selection at the seedlings stage	[144]
Purple sheaths, husks and culm	Bigenic	*B1 & Pl1*	Dominant		Fixation	Haploid selection before flowering stage	[145]
Haploid induction in maternal inducers	Monogenic	*mtl*/*nld*/*zmpla1*, *zmdmp*	Recessive		Fixation	Required for haploid embryo formation	[3,4,5,13]
HIR of maternal inducers	Polygenic	*qhir2-7*, *zmdmp*	Mostly recessive	<0.1–14.5%	Fixation for high HIR	Determines the efficiency in which haploid seeds are created	[11,12,13,15,25]
HIR of paternal inducers	Monogenic	*ig1*	Recessive	0.0–6.0%	Fixation for high HIR	Determines the efficiency in which haploid seeds are created	[2,7,9]
Plant height	Polygenic	>40. E.g.: *Br2*, *D3*, *D8*, *D9*, *na1* and more	Additive, dominant & recessive	0.7–10.4 m	Depend on the method of haploid seed production	Influence the performance on isolation fields and ergonomics in induction nurseries	[106,107,108,109,110,111,112,113]
Tassel size	Polygenic	>24. E.g.: *ba1*, *baf1*, *bif2*, *fea2*, *ra1*, *ra2*, *ra3*, *td1*, *tsh4*, *ub2*, *ub3*, *zfl1*, *zfl2* and more	Mostly recessive	21.1–53.8 cm	Higher values are better	Tassel size influences pollen production, which is important to ensure good seed set in cross-pollinations	[107,109,118,119,120,121,122,123,124,125,126,127,132,146,147]
Seed set in self and cross-pollinations	Polygenic	*ra1*, *ra2*, *ra3*, *ba1*, *bif2*, *bd1*, *bt2*, *fea2*, *ids1*, *td1*	Mostly recessive	0–1348 seeds	Higher values are better	High seeds set in self and cross-pollinations decrease maintenance and DH line production costs, respectively	[118,120,121,122,123,125,131,132,133,134,135,136]
Lodging	Polygenic	*bmr*, *bk2*, *gl1*, *gl15*, *tp1*, *tp2* and more	Mostly recessive	0–100%	Higher resistance is better	Lodged plants may produce lower seed set in isolation fields and reduce ergonomics in induction nurseries	[105,148,149]
Oil content	Polygenic	*lec1*, *DGAT1-2*, *OBAP1*, *WRI1*	Mainly additive	1.7–27.2%	Higher values are better	Higher oil content improves the accuracy of automated discrimination of haploid and diploid seeds	[95,96,97,98,150,151,152,153,154]

**Table 2 plants-09-00614-t002:** Advantages and disadvantages of three inducer variety types.

Trait/Inducer Variety Types	Inbred	Synthetic *	Hybrid
Easiness of production	High	Medium	Low
Suitability for OC-based discrimination	High	Medium	Low
Uniformity and stability	High	Medium	High
Length of pollen shed	Smaller	Higher	Smaller
Pollen yield	Low	Medium	High
Disease tolerance	Low	Medium	High
Performance on isolation fields	Low	Medium	High
Ergonomics for hand-pollinations	High	Medium	Low

* Conclusions for synthetic inducers assumed that they were derived from the cross of two moderately related lines. In case synthetic induces are created by crossing two or more genetic dissimilar parents, their advantages and disadvantages approximate those of hybrid inducers.

**Table 3 plants-09-00614-t003:** Traits of the exotic inducer and noninducer parents assumed for the comparison of the different approaches for inducer development.

Exotic Inducer	Adapted Inbred Lines
*R1-nj* (purple embryo marker)	*r1-nj*
*Pl1* (red root marker)	*pl1*
*mtl*	*MTL*
*zmdmp*	*ZmDMP*
Poor agronomic performance	Good agronomic performance
Poor environmental adaptation	Good environmental adaptation

**Table 4 plants-09-00614-t004:** Selection steps in haploid inducer development.

Generation	Selection Applied	Resulting Breeding Lines
Parental	Selection of elite inbred lines for crossing with exotic inducer	F_1_
F_1_	Discard of F_1_ families with undesirable characteristics	F_2_
F_2_	Discard F_2_ seeds lacking the purple embryo pigmentation. With MAS, fix *mtl* and discard *pl1*/*pl1* or *ZmDMP*/*ZmDMP* genotypes.	F_3_ fixed for *mtl*
F_3_	Fix *R1-nj* by harvesting ears of F_2_ plants that only have colored seeds. Fix *Pl1* and *zmdmp* with MAS. The HIR of the selected genotypes can be evaluated through crosses with one donor	F_4_ fixed for *mtl*, *R1-nj*, *Pl1* and *zmdmp*
F_4,5,6…_	Phenotypic and/or genotypic selection for polygenic traits of importance to inducers	F_5,6,7…_

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
