# Peer review of "Breeding Maize Maternal Haploid Inducers"

_plants, 2020, doi:10.3390/plants9050614_

Round 1

Reviewer 1 Report

The manuscript by Trentin et al provides a well written and timely review of double haploid technology that is a useful tool for maize breeding and includes a useful how to guide and tips that will enable newcomers to the field to rapidly grasp and use this technology in their breeding programs.

I do however have a few minor suggestions for the manuscript

1)lines 156-162: A little more description of the phenotypes of the lesion mimic mutants would be helpful and the transition from the section on seed weight is a bit abrupt.

2) line 175. What is the colchicino treatment for? this is mentioned but no further information is given. Please clarify or remove.

3) line 211. You mention that specialized microscopes are needed for GFP but fail to take into account that NMR used for the high oil determination is also a highly specialized instrument.

4) line 373 please clarify or reference why old germplasm would be more susceptible to ear rots.

5) table 1, lesion mimics and GFP are mentioned in the text as screens for haploids but not included in the table.

6) line 454 define GS at first use.

7) It is a general feeling that the manuscript ends rather abruptly, I would recommend adding a final synthesizing conclusion paragraph.

Reviewer 2 Report

This is a good review with a new angle: how to breed for haploid inducers without a lot of the modern / commercial tools and scale. 

Small critiques are:

  • use of italics consistently throughout with gene names
  • line 230-231: membranes of sperm cells, not pollen grains
  • line 237: explain two synergid cells and the recruitment of a second pollen tube by the persistent synergid after initial single fertilization event
  • Table 1: why is breeding goal 50% for DMP /qhir8. It should be fixation for high HIR
  • Table 1 / inducer breeding section. For WS14: why did ig1 help, if it is a paternal inducer - can you explain why it contributes to higher maternal HIR / or if not, why it was useful?
  • line 401: mention that inbred inducers can have major issues with seed set depression over time
  • Table 3: for ZmDMP - actually the SNP mutation that was found is present in many adapted inbred lines, so it is not clear if ZmDMP or Zmdmp is present in modern adapted varieties. The recent publication from Liu, et al. on DMP does not explain or explore this yet.

Reviewer 3 Report

In general, the authors well described the history of maize maternal haploid inducers, target traits, procedures and strategies for inducer development. As a review manuscript, it included all core papers related to maize haploid inducers, and covered the latest advantages in this field. I have some comments as follows:

In general, I would suggest “3.1.5. Haploid induction” to be an independent section, put it after the section of “2. Status quo of inducer development”, and rename it as “3. Genetic basis and putative biological mechanisms of haploid induction”. Then, put the original section 4 (4. Inducer variety types) before section 3 (3. Breeding goals in developing maize maternal haploid inducers), and call them section 4 and 5, respectively.

Minor comments:

Lines 10-11: The sentence “Self-pollination of haploid plants results in completely homozygous progeny, shortening breeding cycles and increasing genetic gain” might be confusing for two reasons. First, generally the haploid plant itself cannot do self-pollination except a special situation of spontaneous doubling. Secondly, how genetic gain can be increased by doubling haploid plants? If you want to say genetic gain can be potentially improved by shortening the length of breeding cycle interval (L), which might be duplicated with “shortening breeding cycles”.

Line 74: WS14 was developed from the cross of W23ig * Stock6. W23ig is different from inbred line W23.

Lines 70-97: here the authors used three paragraphs to describe the breeding history of maternal haploid inducers. The first paragraph is very well described following a sequential manner. However, the logic behind paragraph 2 (lines 70-82) and paragraph 3 (lines 83-97) was not as clear as the first paragraph. It seems more logical that describe all German inducers (like UH400, UH600, UH601 et al.) in the second paragraph rather than in the third paragraph, because those German inducers are the essential founders of Chinese and CIMMYT inducers. Tropical inducers were also developed in Brazil, it would be good if the authors can include Brazilian breeders’ contribution.

Line 102: the statement “Molecular markers have not been applied in inducer development”, which is not true, see Molecular Breeding 34: 1147–1158. This is one case I know using marker-assisted selection(MAS) for inducer development. For development of CIMMYT inducers, they also used markers because the markers of the major QTL of HI have been published as early as in 2013 (Theor. Appl. Genet. 126: 1713–1720). Considering that two important genes of HI on chromosome 1 and 9 have been cloned, I expect there should be more inducer breeding programs recently using molecular markers.

Lines 150-175: regarding using morphological traits to distinguish haploids from diploids, liguleless phenotype (Genetics 202: 1267–1276), guard cell length (https://doi.org/10.1002/csc2.20004), etc, are also used in breeding practice and genetic research (liguleless test is commonly used in Germany and CIMMYT).

Line 185: The authors said “Melchinger et al. [24,25] reported that diploid embryos had OC levels below the average of the parents”, I am wondering which table/figure from Melchinger papers supported this statement? Here I would expect to see something like OC of haploid embryos is significantly lower than that of diploids, which is the theory of using OC to identify haploids.

Line 214: section "3.1.5. Haploid induction" was not really talking about marker traits, so it's not suitable to put it under "3.1. Marker traits". I suggested to consider it as an independent section (see above).

Line 283: in the section of “3.2.1. Plant height and lodging tolerance”, the authors talked about hybrid inducers. I think it would be better to introduce hybrid inducers beforehand as I suggested switch the sequence of sections 3 and 4.

Line 322-328: regarding how to effectively measure tassel size in maize, It might worth checking whether the modern high throughput phenotyping technology(HTP) e.g. using a drone could provide a better solution.

Line 396: “differences in OC of inbreds used to form a population”, what does this mean?

Line 433: I suggested to change “5. Breeding approaches for inducer development” to “Breeding procedures and strategies for inducer development”
